# Microstrip to Slot-Line-Fed Microstrip Patch Antenna with Radiation Pattern Diversity for X-Band Application

**Jiwan Ghimire** [ID], **Daud Khan and Dong-You Choi** *[ID]

Department of Information and Communication Engineering, Chosun University,
Gwangju 61452, Republic of Korea; ghimire@chosun.kr (J.G.); daudkhalil700@chosun.kr (D.K.)
* Correspondence: dychoi@chosun.ac.kr

**Abstract:** This paper proposes a planar wideband microstrip feedline to a slot-line-based patch antenna for high-frequency pattern diversity applications. The antenna consists of two adjacent rectangular patches separated by the ground slots, with a directive patch slot along the edge of the substrate. A compact common feedline-to-slot configuration is used to miniaturize the antenna. The antenna is designed in stages, starting with an in-phase feedline, followed by a slot line structure, with two radiating patches on top and a director on the side. This creates radiation diversity with directive radiation patterns. The antenna was fabricated and analyzed inside a far-field anechoic chamber. The experimental results validate the return loss, gain, and radiation performance. The measured results of the antenna within a frequency range of 8.5 to 11 GHz show good agreement with the simulation. The antenna has a maximum gain of 9.2 dBi and has the potential to be beneficial for beam steering and X-band applications due to its low profile, broad bandwidth, high gain, and good directivity.

**Keywords:** wide-band antenna; X-band; microstrip patch; slot antenna; beam steering

## 1. Introduction

An antenna with reconfigurable frequency and polarization is becoming more and more essential with the modernization of intelligent devices sensors for use in mobile devices [1], cancer detection [2], wireless devices [3,4], and prospective possibilities for bidirectional communication that can facilitate the integration of several radio networks into specific applications [5–7]. As a result, the design of reconfigurable antennas has gradually received increasing attention and effort, notably in microstrip technology. The frequency, radiation pattern, polarization, or combinations of these are basic antenna characteristics that can be reconfigured [8–13]. To reduce the interference and power consumption, reconfigurable antennas can minimize the signal distortion brought on by multipath fading and various interferences in wireless communication systems [14,15]. In a noisy environment, a null beam of radiation can be targeted at unwanted users to reduce interference, improve security, and minimize noise sources. Pattern-reconfigurable antennas can also adjust the primary beam of the radiation pattern toward the targeted users, which helps to better conserve energy by more efficiently delivering the signal. Traditionally, phased array antennas have been used to perform pattern reconfigurability [16,17]. However, their large size and costly design process have limited their use. Recently, there has been growing interest in pattern-reconfigurable antennas with simple and economical design processes for applications in wireless, military, radar, and satellite communications [18,19]. Various techniques have been proposed to achieve reconfigurability [20]. Switches like radiofrequency micro-electromechanical systems (RF-MEMS), Varactor Diodes, and p-i-n diodes are excellent reconfiguration means of switching technology offering quick switching and reconfigurability in terms of the frequency, bandwidth, radiation pattern, polarization, and front-to-back ratio [21–23]. Liquid material-based frequency and pattern reconfigurable

antennas were proposed in [24,25]. However, their complex structures, parasitic effects on the antenna performance, significant diode demands, switch and biasing circuitry requirements, interference, and lumped elements all led to an increase in the weight, power, and cost. Additionally, developing a frequency-tunable radiation-reconfigurable antenna with the use of passive elements in a limited size is still a challenging task [26,27].

In this paper, we propose a frequency-tunable pattern-reconfigurable antenna. A compact common feed-to-slot line configuration is used. The two radiating patches are printed on top of the Fr4 substrate ($\varepsilon_r$ = 4.4), with directors on the sides, and a slot in the ground plane. As the frequency changes, a relative phase difference occurs in the two-radiating patches of the antenna along the length and width of the patch, which changes the direction of the electromagnetic radiation. This results in pattern reconfigurability. Passive elements that require high power requirements need to be minimized as much as possible, as they can degrade the antenna radiation and efficiency. Therefore, we present a frequency-tunable high-gain radiation pattern-reconfiguration antenna that does not require these power-consuming devices. The antennas are fabricated and analyzed inside a far-field anechoic chamber. The analysis of the antenna is conducted via a simulation using the Ansoft High-Frequency Structure Simulator (HFSS), a commercial electromagnetic simulator. The antenna, which is 18 mm × 28 mm × 0.8 mm in size, was designed on an Fr4 substrate with a 50 Ω microstrip feedline-to-slot-line and slot-line-to-radiating patch power transition. The slot line and radiating patch are separated from the ground plane by the height of the substrate.

## 2. Antenna Design

Figure 1 shows the geometry of the antenna, along with its dimensions and parameters, which are listed in Table 1. The antenna consists of a microstrip feedline of width $F_W$ and length $F_L$, which connects to a radial stub of radius $F_R$. The etched slot of width $S_W$ is of length $S_L + F_W$, and it is etched on the ground plane perpendicularly to the feeding section, where its end is terminated by the circular ground slot of radius $G_R$. The two patches of length and width $P_L \times P_W$ are placed above the ground plane side by side, separated by the distance of slot width. The director of length $D_L$ and width $D_W$ is placed in front of the two rectangular patches so that the radiation becomes more directive at a higher frequency in the X-band. The simulated return loss of the antenna with and without the patches and director is shown in Figure 2a. From the figure, we can see that the patches not only assist in electromagnetic wave radiation but also contribute to the impedance matching of the antenna system. The first and second resonance frequencies of the fabricated antenna can be estimated by using the following equations, as given in [28].

$$f_n = \frac{c}{L_n \sqrt{\varepsilon_r}}, \tag{1}$$

where $c$ is the velocity of light, and $L_n$ is the effective length of the rectangular radiating patch, given that for n = 1, 2, $L_1 = P_L + L_{PF} + F_W/2$, and $L_2 = \sqrt{(P_W^2 + P_L^2)} + S_W$, and $\varepsilon_r$ is the relative permittivity. The design parameters of the proposed antenna are based on the guided wavelength, λg. By varying this base parameter, we can change the dimensions of the antenna and tune its impedance bandwidth. As shown in Figure 2b, when the guided wavelength parameter is varied from 18 mm to 27 mm, the impedance bandwidth of the antenna changes within an ultra-wideband region covering different narrowband regions. This tunning flexibility allows us to adapt the required bandwidth of the antenna by tuning the parameters. Using this flexibility, the antenna is tuned to the bandwidth near the X-band region by setting the guided wavelength (λg) parametric value to 20 mm.

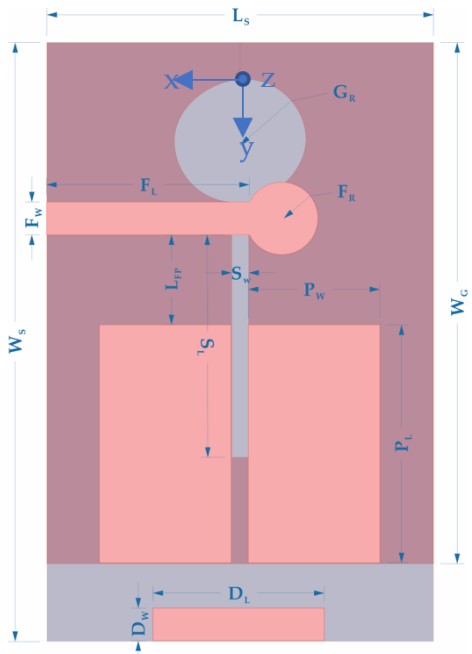

**Figure 1.** Geometry of the proposed fabricated antenna.

**Table 1.** Geometrical parameters of the proposed antenna.

| Parameter | Value (mm) | Parameter | Value (mm) |
|---|---|---|---|
| $L_S$ | $S_L + \lambda g/4 + F_W/2 + S_W \times 2.5$ | $F_W$ | 1.54 |
| $W_s$ | 28 | $F_R$ | $\lambda g/8 - S_W$ |
| $W_G$ | 24.5 | $G_R$ | $\lambda g/8 + S_W/2$ |
| $F_L$ | $L_S/2 + S_W/2$ | $S_W$ | 0.8 |
| $P_L$ | $S_L/2 + \lambda g/4 + F_W/4 + S_W \times 0.75$ | $D_W$ | $F_W$ |
| $P_W$ | $S_L/2 + F_W/4 + S_W \times 0.75$ | $D_L$ | 8 |
| $S_L$ | $\lambda g/2 + S_W/2$ | $\lambda g$ | 20 |
| $L_{PF}$ | $\lambda g/4 + S_W - F_W$ | | |

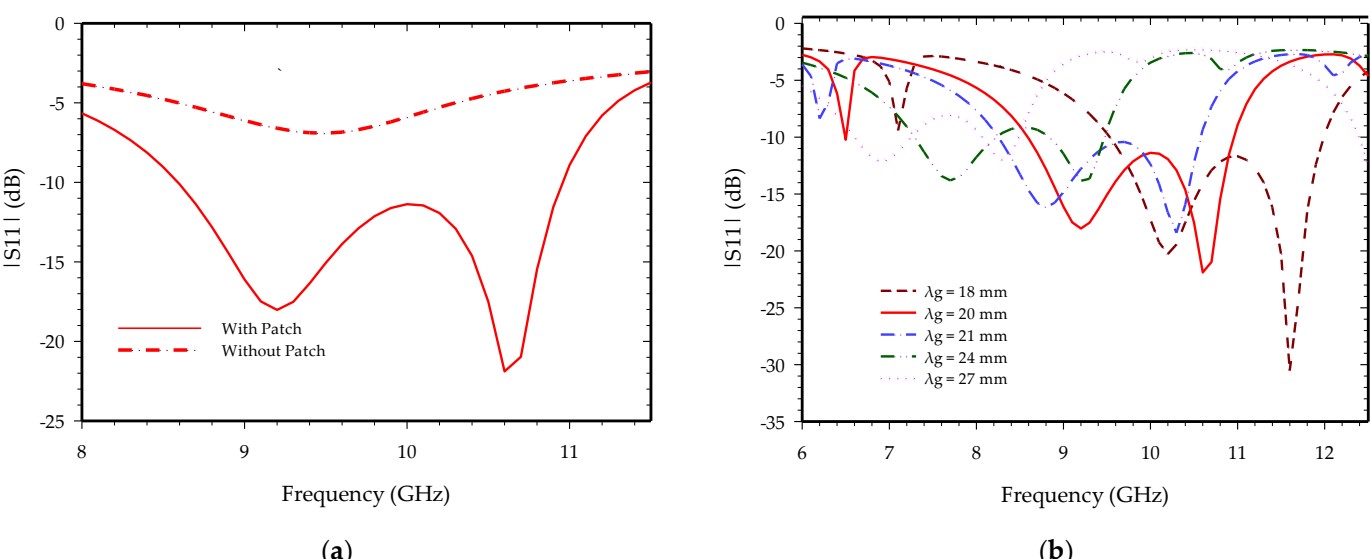

**Figure 2.** Simulated return loss: (**a**) the antenna with and without radiating patches; (**b**) at different parametric values of guided wavelength parameter λg.

## 3. Simulated and Measured Results

The simulation and measured results are shown in Figure 3. The figure shows the impedance bandwidth, where |S11| < −10 dB, varies from 8.5 to 11 GHz. The resonance frequencies are at around 8.9 and 10.5 GHz. The slot length, followed by the patch length, contributes to the lower-frequency resonance, whereas the diagonal length of the patch contributes to the upper-frequency resonance. The simulated and measured return loss vary significantly, which can be attributed to the connection losses, faulty soldering, and fabrication flaws.

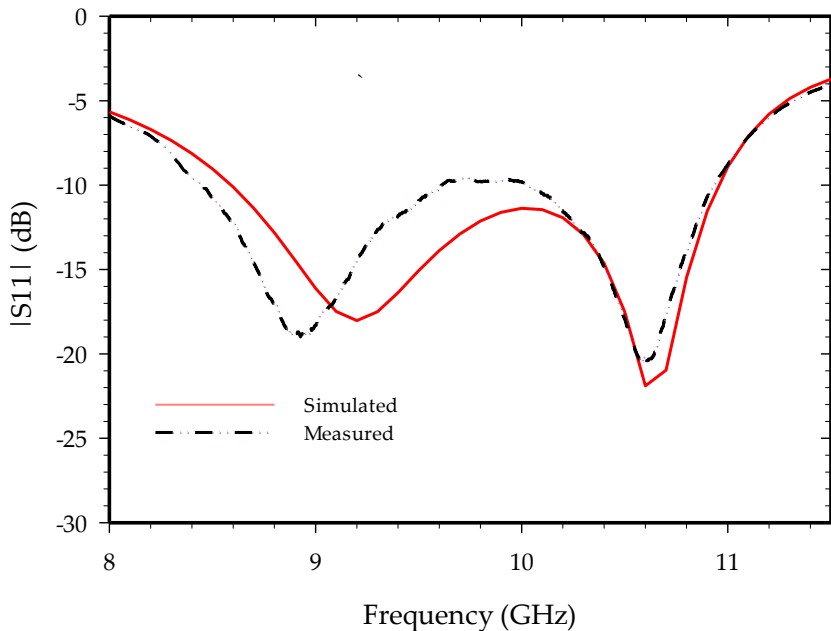

**Figure 3.** Simulated and measured S-parameters of the proposed antenna.

The measured 3D radiation pattern at 8.8, 9.6, and 10.6 GHz frequencies are shown in Figure 4 for both the patch antennas, with and without a director. The far-field radiation patterns are directive and change the direction of radiation with the change in frequencies. The insertion of a director slot at the patch antenna tends to make the radiation pattern more directive at higher frequencies towards the azimuthal plane, which can be seen in the radiation plot of Figure 4f. In contrast, the radiation at the elevation plane is more directive for a patch antenna without a director at a lower frequency, as shown in Figure 4a. Around the central frequency of the proposed antenna at 9.6 GHz, the direction of radiation is in between the elevation and the azimuth plane. The measurements of the antennas were performed in an anechoic chamber, as shown in Figure 5a. Figure 5b shows the total realized gain plot of antennas with and without a director, as well as the fabricated antenna at the side of the graph. The realized gain of both the antennas is within 2 to 9.2 dBi in the X-band region. The gain variation of the antenna with a director is nearly identical to that of the antenna without a director, except at the upper frequency in the X-band. At this frequency, the director directs the radiation towards the azimuthal plane, which increases the directivity and gain of the antenna. Furthermore, the antenna without a director radiates in all directions, both in the elevation and azimuth planes, which is likely to decrease the peak gain. The surface current distribution at 8.8 and 10.6 GHz frequencies is depicted in Figure 6. The surface current concentration is high at the radiating patch along the horizontal edge at 8.8 GHz (Figure 6a,b), whereas on the vertical edge of the patch and director, the current density seems to be more at the antenna operating frequency of 10.6 GHz (Figure 6c,d). The surface current at the horizontal edge of the patch leads to the direction of radiation at the elevation plane, which can be seen in the radiation pattern of Figure 4a,b, while the surface current concentration at the vertical edge of the patch

tends to radiate in the direction of the azimuthal plane (Figure 4e,f). The in-phase current between the director and the radiating patches likely tends to the radiation pattern being more directive at the azimuthal plane.

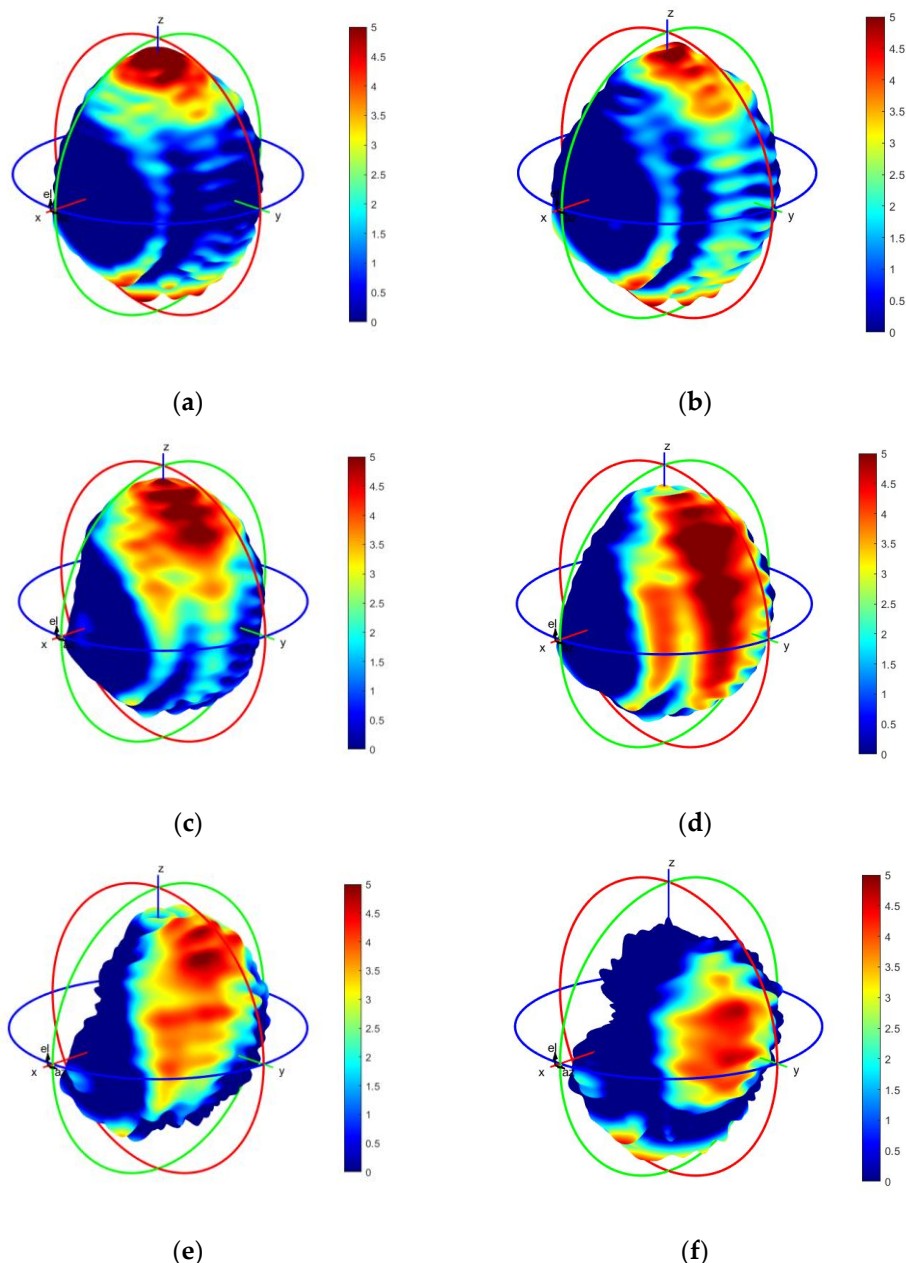

(a) (b)

(c) (d)

(e) (f)

**Figure 4.** Measured far-field 3D radiation pattern for antennas without a director (**a**,**c**,**e**) and with a director (**b**,**d**,**f**) at frequencies 8.8, 9.6, and 10.6 GHz respectively. The color bar represents the total gain of the antenna in the dBi scale.

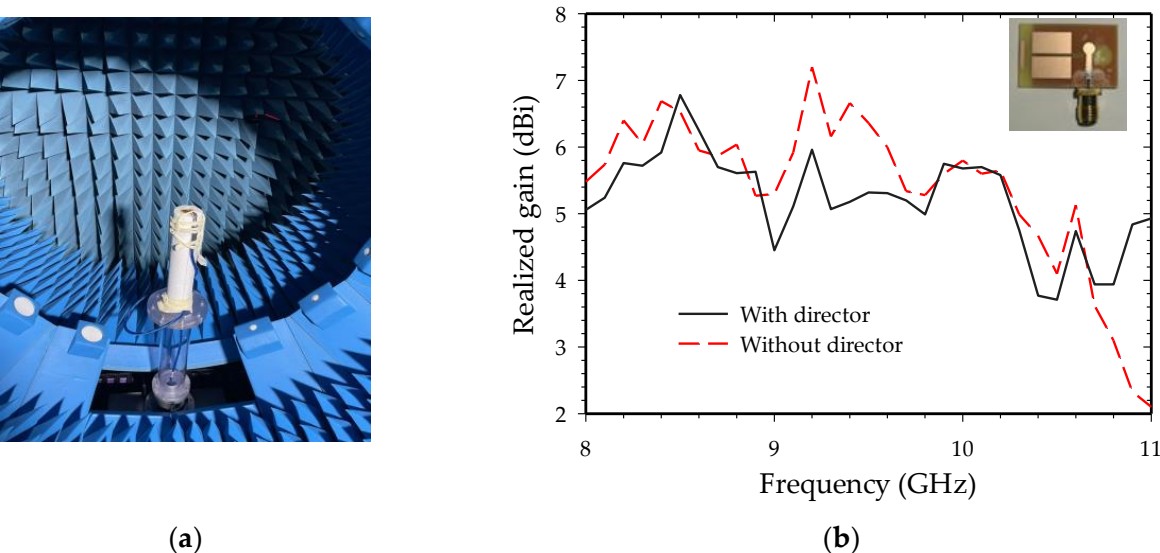

(**a**)   (**b**)

**Figure 5.** Measurement setup and the gain of the proposed antennas: (**a**) radiation pattern measurement in the anechoic chamber; (**b**) measured realized gain.

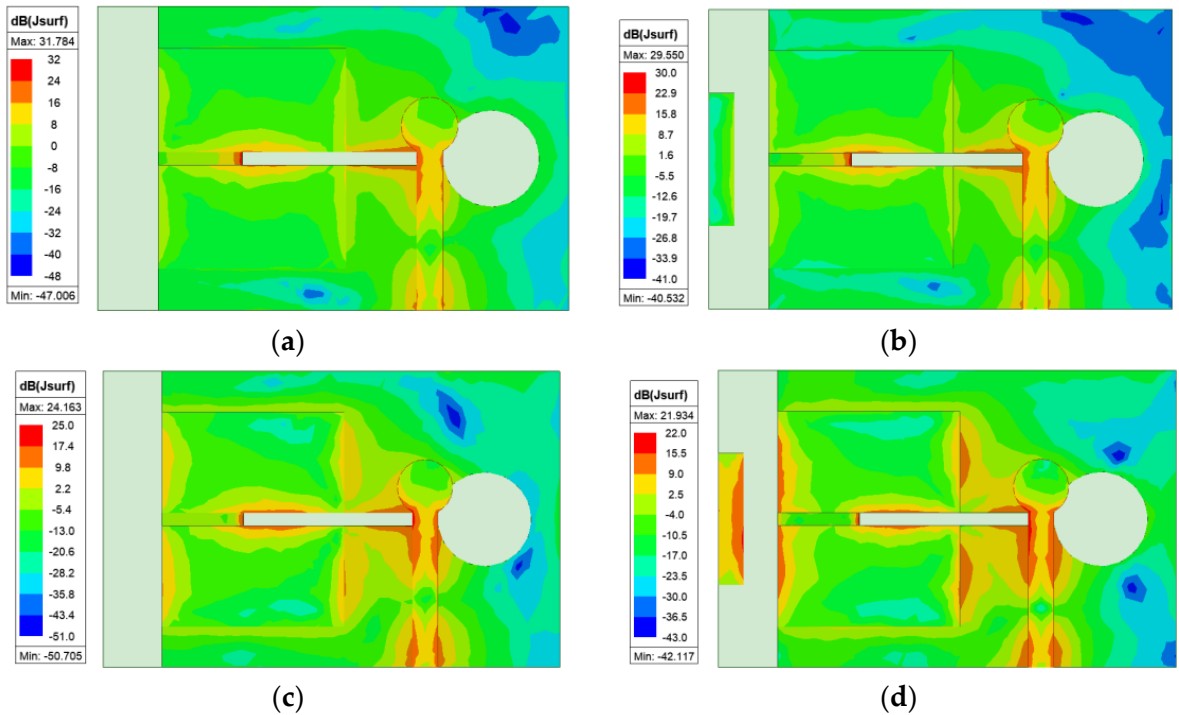

**Figure 6.** The simulated surface current distribution of the proposed antennas at 8.8 GHz (**a**,**b**) and 10.6 GHz (**c**,**d**) for both antennae with and without a director, respectively.

## 4. Conclusions

A wideband microstrip feedline to the slot line and the slot line to two adjacent radiating patch-based antennas intended to cover most of the X-band signals was proposed in this paper. The antenna's frequency covers 8.5 to 11 GHz and has a maximum gain of 9.2 dBi. The feedline-to-slot-line and slot-line-to-radiating patch power transitions take place, where the directive patch slot along the edge of the substrate helps to radiate the antenna at the azimuth plane at the higher frequency end, whereas the antenna radiation is directive at the elevation plane at the lower frequency end of the X band. This allows the designed antennas to radiate into the desired directions, so that a null beam of radiation

can be targeted at unwanted users to reduce interference, improve security, and minimize noise sources. Additionally, the use of power-hungry switching devices, such as diodes and resistors, in an antenna can significantly reduce its power efficiency. The fabricated antenna was analyzed inside a far-field anechoic chamber. The experimental results validated the return loss, gain, and radiation performance. However, one limitation of this antenna design is that the antenna radiation is confined between the elevation and azimuthal planes. The measured results of the antenna show good agreement with the simulation, which has the advantages of a low profile, broad bandwidth, high gain, beam steering, and good directivity. Additionally, the antenna's bandwidth can be changed by tuning its parameter, which gives it the flexibility to operate at different narrowband frequency applications. These advantages make the antenna a good potential candidate for the X-band (8–12 GHz) applications, as well as other smart radar-based systems, wireless computer networks, medical imaging, and industrial applications.

**Author Contributions:** Conceptualization, J.G.; methodology and simulation, J.G.; writing—original draft preparation, J.G.; writing—review and editing, D.K. and J.G.; supervision, D.-Y.C.; funding acquisition, D.-Y.C. All authors have read and agreed to the published version of the manuscript.

**Funding:** This study was supported by research fund from Chosun University, 2023.

**Data Availability Statement:** Not applicable.

**Conflicts of Interest:** The authors declare no conflict of interest.

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
