# Peer review of "Microstrip to Slot-Line-Fed Microstrip Patch Antenna with Radiation Pattern Diversity for X-Band Application"

_electronics, doi:10.3390/electronics12173672_

Round 1

Reviewer 1 Report

The following suggestions should be addressed:

1. Fig. 1 - the length scale is missing

2. Fig. 1, cont'd - quantities shown in the image of the fabricated antena need to be defined in the caption and their numerical values need to be specified in the figure caption

3. Fig. 2 - measured data should be presented by dots of some kind, not by the dashed line; otherwise, the readers can't reliably establish the values

4. Fig. 4 - the length scale is missing from the picture

5. Fig. 5 - length scale is missing here, too.

6. About conclusions - there should be a more broad discussion (say, 2 paragraphs) of the implications and limitations of this research. 

no comments on English

Author Response

We appreciate the valuable suggestions provided by the reviewer. The dimensions of the designed antenna are outlined in Table 1. Furthermore, all figures within the paper feature the identical dimensions as the proposed antenna. For scale reference, we can compaire  the SMA connector as standard scale. The conclusion section elaboration on the implications and limitations of the design has been included. 

Reviewer 2 Report

It is good paper where particular structure is described. It is conidered in detailes by simulation and measurement. Good consistence is shown. Some additional details of resons to desing of partulular elements of structure under consideration namely in such forms and of such sizes can be usefull for an interested reader.

Some corrections:

Line - comment

9 - Microstrip->microstrip

42 - lumpy->lumped

48 - two-radiating patch -> two radiating patches

50 - as->. As

Section 2 - all variables in text and Table 1 must be intallic

69 - resonating->resonance

72 - Where->where (and move to left)

77 - parameter->parameters

81 - S11->|S11|

In title and vertical axes of Fig. 2 - S-parameters->->|S11|

153, 154 - Remove missing Enter?

Author Response

We greatly appreciate the valuable suggestions provided by the reviewer. We have incorporated supporting information into the introduction and antenna design section and have also addressed necessary corrections.

Reviewer 3 Report

The text is written very briefly. Parts such as fabrication method and details such as impedance and comparison of work with similar works should be mentioned. Please make the text more complete and resubmit. Figure 5 is briefly explained in case it is important for this design. It is better to write the connection between figure 4, 3 and 5 in the text.

Author Response

We are greatly thankful for the valuable suggestions of the reviewers. The valuable comments have been revised and elaborated in this short form of manuscript (communication). Fabrication methods are discussed, and a comparison is made in terms of lightness, weight, and power consumption. We also discuss how simple an antenna can be made without the need for passive devices. A further brief explanation is made in terms of design, graphs, and plots.

Reviewer 4 Report

This paper presents a planar wideband Microstrip feedline to slot line-based patch antenna for high-frequency pattern diversity purposes. However, some descriptions are not clear. Some revisions are necessary in the manuscript.

1. Please further elaborate the effectiveness of the method from the theoretical level.

2. Please further elaborate on the innovation and significance of the article.

3. Please add some mathematical models.

4. Please further explain whether the parameter values in Table 1 are universal.

5. In the paper, authors have focused on frequency and polarization reconfigurable in microstrip technology. Parameter impact on the system needs to be analyzed to indicate advantages of your work, which can refer to

[a] IEEE Transactions on Industrial Informatics, vol. 18, no. 2, pp. 835-846, 2022

[b] IEEE Transactions on Industrial Informatics, 2023, DOI: 10.1109/TII.2023.3241682

[c] IEEE Transactions on Circuits and Systems II: Express Briefs, vol. 69, no. 3, pp. 1029-1033, 2022

[d] IEEE Antennas and Wireless Propagation Letters, vol. 22, no. 2, pp. 367-371, 2023

A proofreading is needed.

Author Response

We are greatly thankful for the reviewers' valuable suggestions. The valuable comments have been revised and elaborated in the manuscripts.

Round 2

Reviewer 3 Report

I accept this manuscript to be published

Reviewer 4 Report

No further comments.